# Multistable Shape from Shading Emerges from Patch Diffusion

**Xinran Nicole Han**
Harvard University
xinranhan@g.harvard.edu

**Todd Zickler**
Harvard University
zickler@seas.harvard.edu

**Ko Nishino**
Kyoto University
kon@i.kyoto-u.ac.jp

## Abstract

Models for inferring monocular shape of surfaces with diffuse reflection—shape from shading—ought to produce distributions of outputs, because there are fundamental mathematical ambiguities of both continuous (e.g., bas-relief) and discrete (e.g., convex/concave) types that are also experienced by humans. Yet, the outputs of current models are limited to point estimates or tight distributions around single modes, which prevent them from capturing these effects. We introduce a model that reconstructs a multimodal distribution of shapes from a single shading image, which aligns with the human experience of multistable perception. We train a small denoising diffusion process to generate surface normal fields from $16 \times 16$ patches of synthetic images of everyday 3D objects. We deploy this model patch-wise at multiple scales, with guidance from inter-patch shape consistency constraints. Despite its relatively small parameter count and predominantly bottom-up structure, we show that multistable shape explanations emerge from this model for ambiguous test images that humans experience as being multistable. At the same time, the model produces veridical shape estimates for object-like images that include distinctive occluding contours and appear less ambiguous. This may inspire new architectures for stochastic 3D shape perception that are more efficient and better aligned with human experience.

## 1 Introduction

From chiaroscuro in Renaissance paintings to the interplay of light and dark in Ansel Adams' photographs, humans are masters at perceiving three-dimensional shape from variations of image intensity—shading—from a single image alone. Even though our visual experience is dominated by everyday objects, our perception of shape from shading generalizes to many synthetic "non-ecological" images invented by vision scientists. Some of these images have ambiguous (e.g., convex/concave) interpretations and lead to multistable perceptions, where one's impression of 3D shape alternates between two or more competing explanations. Figure 1 shows an example adapted from [31], which is alternately interpreted as an indentation or a protrusion. Both explanations are physically correct because the same image can be generated from either shape under different lighting conditions.

How can a computational model acquire this human ability to capture multiple underlying shape explanations? An algorithmic suggestion comes from Marr's principle of least commitment [34, 35], which requires not doing anything that may later have to be undone. But this is in contrast to many computer vision methods for shape from shading, including SIRFS [4] and recent neural feed-forward models [53, 56], which are deterministic and produce a single, best estimate of shape. These types of models commit to one explanation based on priors that are either designed or learned from a dataset, and they cannot express multiple interpretations of an ambiguous image. They are unlikely to be good models for the mechanisms that underlie multistable perception.

38th Conference on Neural Information Processing Systems (NeurIPS 2024).

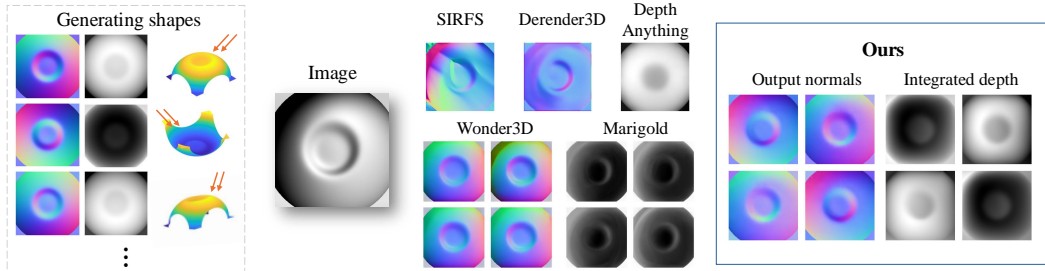

Figure 1: Many shapes (left) can explain the same image (middle) under different lighting, including flattened and tilted versions and convex/concave flips. The concave/convex flip in this example is also perceived by humans, often aided by rotating the image clockwise by 90 degrees. Previous methods for inferring either surface normals (SIRFS [4], Derender3D [53], Wonder3D [33]) or depth (Marigold [27], Depth Anything [56]) produce a single shape estimate or a unimodal distribution. Ours produces a multimodal distribution that matches the perceived flip. (Image adapted from [31].)

Instead, we approach monocular shape inference as a conditional generative process, and inspired by a long history of shape from shading with local patches [18, 25, 9, 29, 54, 20, 17], we present a bottom-up, patch-based diffusion model that can mimic multistable perception for ambiguous images of diffuse shading. Notably, our model is trained using images of familiar object-like shapes and has no prior experience with the ambiguous images that we use for testing. It is built on a small conditional diffusion process that is pre-trained to predict surface normals from $16 \times 16$ image patches. When we apply this patch process at multiple scales with inter-patch shape consistency constraints, and when we coordinate the sampling across patches, the model ends up capturing global ambiguities that are very similar to those experienced by humans.

An important attribute of our model is the way it handles lighting. It builds on the mathematical observation that shape perception can precede lighting inference [30]. It also adheres to the philosophy that inferred lighting cannot, and should not, be precise because of spatially-varying effects like global illumination [50]. Our model achieves these aims by guiding its diffusion sampling process with a very weak constraint on lighting consistency, where each patch nominates a dominant light direction and then all patches enact their own concave/convex flips in response to those nominations.

Another critical aspect of our model is a diffusion sampling process that is coordinated across multiple scales. It involves spatially resampling the normal predictions at intermediate diffusion time steps and then adding noise before resuming the diffusion at the resampled resolutions. Our approach is inspired by previous work that uses a "V-cycle" (fine-coarse-fine) to avoid undesirable local extrema during MRF optimization [36]. Our ablation experiments show that multi-scale sampling is crucial for finding good shape explanations that are globally consistent.

We train our patch diffusion model on images of objects like those in Fig. 2a, and we find experimentally that it can generalize to new objects as well as to images like Fig. 1 which are quite different from the training set and appear multistable to humans. This is in contrast to previous diffusion-based monocular shape models [27, 33] which cannot capture multistability and produce output that is much less diverse. Our model is also extremely efficient, only requiring a small pixel-based diffusion UNet that operates on $16 \times 16$ patches. Our total model weights require only 10MB of storage, much less than the 2-3GB required by some of the previous (and more general) models we compare to.

## 2 Background

### 2.1 Ambiguities in Shape from Shading

Shape from shading is a classic reconstruction problem in computer vision. Since being formulated by Horn in the 1970s [23] there have been many approaches to tackle it, often by assuming diffuse Lambertian shading and uniform lighting from either a single direction (e.g.,[43, 16]) or as low-order spherical harmonics (e.g., [4, 59]). Almost all methods either require the lighting to be known (e.g., for natural illumination [41]) or try to estimate it precisely via inverse rendering during the optimization process [54, 59, 53], and many approaches rely on a set of priors to constrain the possible

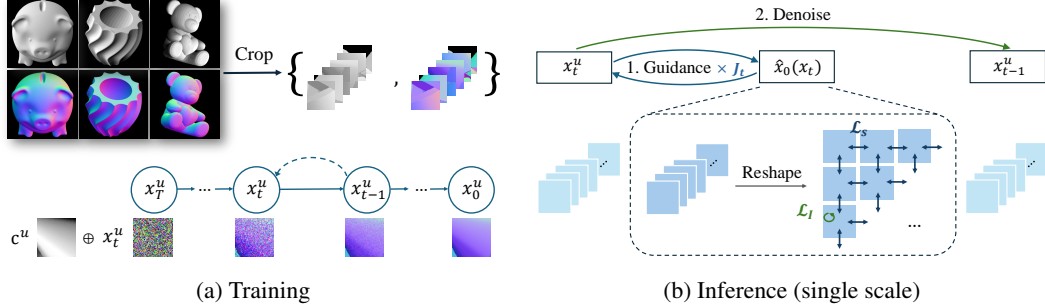

(a) Training                 (b) Inference (single scale)

Figure 2: Training patches are cropped from synthetic images of ordinary diffuse objects, and during training, a small diffusion model learns to denoise the normal field $x_0^u$ for patch $u$ from a random sample $x_T^u$ conditioned on the patch intensities $c^u$. During inference, the model is applied in parallel to non-overlapping patches, with guidance from inter-patch shape-consistency constraints to minimize the curvature smoothness loss $\mathcal{L}_S$ and integrability loss $\mathcal{L}_I$.

search space. For instance, SIRFS [4] uses different lighting priors for natural versus laboratory conditions and a surface normal prior along occluding contours. Recent deep learning approaches like Derender3D [53] have demonstrated impressive results without being limited to Lambertian reflectance, but they similarly rely on priors internalized from their training sets and have trouble generalizing to new conditions.

The main challenge of shape from diffuse shading comes from the many levels of inherent ambiguity. At a single Lambertian point, when lighting is unknown, there is a multi-dimensional manifold of surface orientations and curvatures that are consistent with the spatial derivatives of intensity at the point [29, 20]. Even when lighting and surface albedo are known at a point, there is a cone of possible normal directions. At the level of a quadratic surface patch, when lighting is unknown, there is a discrete four-way ambiguity corresponding to convex, concave, and saddle shapes [54, 29]. At a global level, when lighting and surface albedo are known, ambiguities arise from interpreting the Lambertian shading equation as a PDE (e.g., [6]) or as a system of polynomial equations [13]. And when lighting and albedo are unknown, there is an additional three-parameter global ambiguity that corresponds to flattenings and tiltings of the global shape [5]. Finally, when lighting is unknown, a global shape has a discrete counterpart that corresponds to a global convex/concave flip.

It is important to note that all of these mathematical ambiguities are based on certain idealized models for the image formation process, such as exact Lambertian shading, perfectly uniform albedo, and most commonly, perfectly uniform lighting that ignores global illumination effects such as interreflections and ambient occlusion, which in reality have substantial effects [50]. An advantage of a stochastic, learning-based approach, like the one presented here, is the potential to capture all of these ambiguities as well as others that have not yet been discovered or characterized.

## 2.2 Denoising Diffusion with Guidance

Diffusion probabilistic models [22](DDPM) generate data by iteratively denoising samples from a Gaussian (or other) pre-determined distribution. We build on a conditional denoising diffusion model, where the condition $c$ is a grayscale image patch, and the model is designed to approximate the distribution $q(x_0|c)$ on 3-channel normal maps $x_0$ with a tractable model distribution $p_\theta(x_0|c)$. A 'forward process' adds Gaussian noise to a clean input $x_0$ and is modeled as a Markov chain with Gaussian transitions for timesteps $t = 0, 1, \cdots, T$. Each step in the forward process adds noise according to $q(x_t|x_{t-1}, c) := \mathcal{N}(\sqrt{1 - \beta_t}x_{t-1}, \beta_t\mathbf{I})$, where $\{\beta_t\}$ is a predetermined noise variance schedule. The intermediate noisy input $x_t$ can be written as

$$x_t = \sqrt{\alpha_t}x_0 + \sqrt{1 - \alpha_t}\omega, \quad \omega \sim \mathcal{N}(0, \mathbf{I}), \quad \text{where } \alpha_t := \prod_{s=1}^{t}(1 - \beta_s). \tag{1}$$

The 'reverse process' $q(x_{t-1}|x_t, c)$ is modeled by a learned Gaussian transition $p_\theta(x_{t-1}|x_t, c) := \mathcal{N}(x_{t-1}; \mu_\theta(x_t, t; c), \sigma_t^2\mathbf{I})$. The mean value $\mu_\theta(x_t, t; c)$ can be expressed as a combination of the noisy image $x_t$ and a noise prediction $\epsilon_\theta(x_t, t; c)$ from a learned model. The noise prediction model

$\theta$ can be trained by minimizing the prediction error

$$L(\theta) := \mathbb{E}_{x_0, \omega \sim \mathcal{N}(0, \mathbf{I})} \left[ ||\omega - \epsilon_\theta(x_t, t; c)||_2^2 \right], \tag{2}$$

as shown in [22]. To sample noiseless data using the learned model, we start from an initial random Gaussian noise seed and use the learned denoiser $\epsilon_\theta$ to compute $x_{t-1} = \frac{1}{\sqrt{1-\beta_t}}(x_t - \frac{\beta_t}{\sqrt{1-\alpha_t}}\epsilon_\theta(x_t, t; c)) + \sigma_t z$ iteratively, where $z \sim \mathcal{N}(0, \mathbf{I})$, which is a stochastic process.

The denoising diffusion implicit model (DDIM) shows that the reverse procedure can be made *deterministic* by modeling it as a non-Markovian process with the same forward marginals [48]. This approach helps to accelerate the sampling process by using fewer steps and also provides an estimate of the predicted $\hat{x}_0$ at each timestep with $x_t$. Each denoising step combines noise and a *foreseen* denoised version

$$x_{t-1} = \sqrt{\alpha_{t-1}} f_\theta(x_t, t; c) + \sqrt{1 - \alpha_{t-1}} \epsilon_\theta(x_t, t; c), \tag{3}$$

where

$$f_\theta(x_t, t; c) = \hat{x}_0(x_t) = \frac{x_t - \sqrt{1 - \alpha_t} \epsilon_\theta(x_t, t; c)}{\sqrt{\alpha_t}} \tag{4}$$

is the predicted $\hat{x}_0$ at reverse sampling step $t$.

The DDIM formulation provides a way to 'guide' the process of sampling $x_{t-1}$ from $x_t$ by applying additional constraints or losses to the predicted $\hat{x}_0(x_t)$ at intermediate sampling steps. Previous work has used similar approaches to solve inverse problems [8, 47] or to combine outputs from multiple diffusion models for improved perceptual similarity [32]. Guided denoising is achieved with

$$x_t' = x_t - \eta_t \nabla_{x_t} \mathcal{L}(\hat{x}_0(x_t)), \tag{5}$$

$$x_{t-1} = \sqrt{\alpha_{t-1}} \hat{x}_0(x_t') + \sqrt{1 - \alpha_{t-1}} \epsilon_\theta(x_t, t; c), \tag{6}$$

in each sampling subroutine from time $t$ to $t - 1$, where $\eta_t$ is the (possibly time-dependent) step size of the guided gradient update. The first step (5) can be repeated multiple times before applying the denoising step (6).

## 3 Multiscale Patch Diffusion with Guidance

Consider a surface represented by a differentiable height function $h(x, y)$ over image domain $(x, y)$ viewed by parallel projection from above. The image plane is sampled on a square grid (pixels). Image patches have size $d \times d$ and are indexed by $u$, and we denote them as $c^u \in [0, 1]^{d \times d}$.

Training occurs on patches extracted from images of everyday objects, as depicted in Fig. 2a. For each image patch $c^u$ there is a corresponding patch normal field $x_0^u \in [-1, 1]^{3 \times d \times d}$, whose $(i, j)$th spatial element represents a surface normal vector via

$$\frac{x_0^u(i, j)}{\|x_0^u(i, j)\|} = \frac{(-p_{i,j}, -q_{i,j}, 1)}{\|(-p_{i,j}, -q_{i,j}, 1)\|} = n_{i,j} \in \mathbb{S}^2, \tag{7}$$

where $(p, q) = (\partial h / \partial x, \partial h / \partial y)$ are the surface derivatives. Training proceeds as described in Sec. 2.2, with a dataset of patch tuples $(c^u, x_0^u)$ and a UNet $\epsilon_\theta(x_t^u, t; c^u)$ similar to that in [22] whose four-channel input is the concatenation $c^u \oplus x_t^u$. (Model and training details are in Appendix A.2.)

As depicted in Fig. 2b, inference occurs over images $c$ of size $H \times W$ which are divided into their collections of non-overlapping patches $c^u$. We assume that the underlying global normal field $x_0 \in [-1, 1]^{3 \times H \times W}$ is continuous including at most locations on the seams between the patch normal fields $x_0^u$. We formulate the prediction of global field $x_0$ as reverse conditional diffusion on an undirected, four-connected graph $\mathcal{G}(\mathcal{V}, \mathcal{E})$. Each patch $x_0^u, u \in \mathcal{V}$ is a node, and there are edges $\{u, v\} \in \mathcal{E}$ between pairs of patches that are horizontally or vertically adjacent.

To encourage the patch fields to form a globally coherent prediction, we use guidance as described by Eqs. 5 and 6. Our guidance includes two terms:

$$\mathcal{L}(\hat{x}_0) = \frac{1}{|\mathcal{E}|} \sum_{\{u,v\} \in \mathcal{E}} \mathcal{L}_S(\hat{x}_0^u, \hat{x}_0^v) + \lambda \frac{1}{|\mathcal{V}|} \sum_{v \in \mathcal{V}} \mathcal{L}_I(\hat{x}_0^v), \tag{8}$$

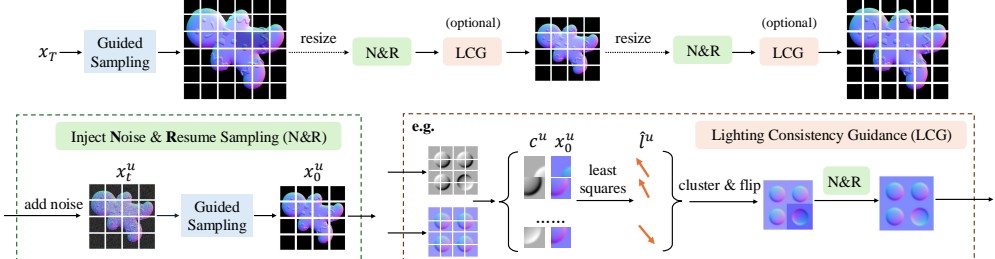

Figure 3: *Top*: Illustration of multiscale sampling across two scales in a fine-coarse-fine "V-cycle", with conditional images omitted for simplicity. In practice, our V-cycle covers more than two scales. *Left*: The $N\&R$ subroutine injects noise to an earlier timestep $0 < t < T$ and then resumes guided sampling (Fig. 2b) at that scale. *Right*: Optional intermediate guidance comes from lighting consistency (LCG), where each patch nominates a dominant light direction and then some patches flip in response to those nominations. Pseudocode is in the appendix.

where $\mathcal{L}_I$ is a within-patch continuity term that encourages integrability of the normal fields over small pixel loops, and $\mathcal{L}_S$ is an inter-patch spatial consistency term that encourages constant curvature across the seams between patches. Hyperparameter $\lambda \in (0, 1]$ controls the relative weighting of the two terms, and $\eta_t \geq 0$ in Eq. 5 determines the overall guidance strength.

The **integrability** term follows Horn and Brooks [24] by penalizing deviation from a discrete approximation to the integrability of surface normals, i.e., $\partial p / \partial y = \partial q / \partial x$, over $2 \times 2$ loops of pixels. We write this as

$$\mathcal{L}_I(\hat{x}_0^u) = \sum_{i,j} (p_{i,j+1} - p_{i+1,j+1} + p_{i,j} - p_{i+1,j} + q_{i,j+1} + q_{i+1,j+1} - q_{i,j} - q_{i+1,j})^2, \quad (9)$$

where the summation is over the $i, j$ grid-indexed pixels in patch $u$, and $p, q$ are the components of $\hat{x}_0^u$ implied by Eq. 7.

The **spatial consistency** term penalizes deviation from constant surface curvature across each seam $\{u, v\} \in \mathcal{E}$ in the direction perpendicular to the seam. Consider four consecutive normals in the perpendicular direction $n_1, n_2, m_1, m_2$ where $n_i$ belong to patch $u$ and $m_i$ belong to patch $v$. We penalize the absolute angular difference between $m_1$ in $v$ and its extrapolated estimate using normals in $u$, i.e., $n_2 + (n_2 - n_1)$. Making this symmetric gives

$$\left\| \cos^{-1} \left( m_1 \cdot \left( n_2 + (n_2 - n_1) \right) \right) \right\| + \left\| \cos^{-1} \left( n_2 \cdot \left( m_1 - (m_2 - m_1) \right) \right) \right\|, \quad (10)$$

which we sum over the length of the seam to obtain $\mathcal{L}_S(\hat{x}_0^u, \hat{x}_0^v)$.

### 3.1 Dominant Global Lighting Constraint

Our guidance so far enforces global coherence, but even globally coherent surfaces can contain regions that independently undergo convex/concave flips without affecting their surround [31]. The top row of Fig. 5 provides a familiar example. One can hide any three of the bumps/dents and then perceive the fourth as being either concave or convex. Yet, when one is allowed to examine the image as a whole and rotate it upside down, instead of perceiving $2^4 = 16$ interpretations, one sees only two: the bottom-right element is a bump (or dent) and the other three are its opposite. This behavior is explained by lighting. If one expects the dominant light direction to be similar everywhere on the surface, the four flips become tied together.

We can incorporate this notion of dominant light consistency into our model using an additional discrete guidance step, as depicted in the bottom right of Fig. 3. This step can be applied to any global sample $\hat{x}_0$ and has three parts: (*i*) patches $\hat{x}_0^u$ independently nominate dominant light directions $\hat{l}^u$; (*ii*) we identify a single direction $\hat{l}$ that is most common among these nominations; and (*iii*) some patches perform a concave/convex flip to become more consistent with $\hat{l}$.

Specifically, each patch $\hat{x}_0^u$ that is not too close to being planar (i.e., that has non-constant $\hat{x}_0^u(i)$) nominates its dominant light direction by computing the least-squares estimate according to shadowless

Lambertian shading:

$$\hat{l}^u = \arg\min_{l \in \mathbb{R}^3} \sum_i \left( c^u(i) - \frac{\langle \hat{x}_0^u(i), l \rangle}{\|\hat{x}_0^u(i)\|} \right)^2 . \tag{11}$$

We create two clusters in the set $\{\hat{l}^u\}$ using $k$-means and choose the center of the majority cluster as the dominant global direction $\hat{l}$. Each patch $u$ in the minority cluster undergoes a concave/convex flip $(p, q) \to (-p, -q)$ by multiplying $(-1)$ with the first two channels of $\hat{x}_0^u$. Since the independent flips can cause discontinuities at patch seams, we always follow this discrete lighting guidance step by an *inject Noise & Resume sampling* (N&R) subroutine, where we add noise to an intermediate timestep via Eq. 1 and resume spatially-guided denoising from that timestep. Pseudocode is provided in Appendix A.1.

This approach to lighting consistency has several advantages. Unlike many previous computational approaches to shape from shading, it does not assume the lighting to be known beforehand. Nor does it require the lighting to be exactly spatially uniform across the surface, which provides some resilience to global illumination effects. It imposes no prior on the dominant light direction (e.g., 'lighting from above'), but one can imagine extending it to do so. And because our patch-based framework can be applied with or without lighting consistency guidance (see Appendix A.3), it may provide a mechanism in the future for modeling the way in which humans selectively enforce lighting consistency across an image [7, 40].

## 3.2   Multiscale Optimization

Since each local image patch can be explained by either concave or convex shapes, the terms in the spatial guidance energy (Eq. 8) are multimodal, and finding a global minimum is computationally difficult. Our experiments, like the one in Fig. 4, show that optimization at a single scale with random initial noise and gradient-descent guidance often gets trapped in poor local minima. To overcome this, and also to fully leverage shading information from various spatial frequencies and scales, we draw inspiration from work on Markov random fields [36] and introduce a multiscale optimization scheme. This is possible because our patch diffusion UNet and guidance can be applied to any resampled resolution $(sH) \times (sW)$ that is divisible by patch size $d$.

Our multiscale optimization occurs in a "V-cycle", a sequence of fine-coarse-fine resolutions. We begin by applying guided denoising at the highest image resolution. Then, we downsample the predicted global field to a lower resolution before injecting noise and resuming reverse sampling $(N\&R)$ at that lower resolution. As depicted in the left of Fig. 3, this has the effect of generating a random sample at the lower resolution that is informed by a previous sample at the higher resolution. A similar process occurs when going from coarse to fine, but with the global field being upsampled before applying the $N\&R$ subroutine.

To further reduce discontinuities at seams, we find it helpful to store and fuse the global field estimates from the final few resolutions of the fine-coarse-fine cycle. We do this by computing their $(p, q)$ fields via Eq.7, resampling them to the highest resolution and averaging them, and then converting the average back to a normal field.

## 4   Experimental Results

The input to our patch UNet is the concatenation of $c^u$ and $x_t^u$. It has 4 channels and spatial dimension $d \times d$ with $d = 16$. We train it using patches of size $d \times d$ extracted from rendered images of the 3D objects in [26] curated from Adobe Stock. We use Lambertian shading from random light directions, with a random albedo in $[0.5, 1]$ and without cast shadows or global illumination effects. Our dataset contains around 8000 images $(256 \times 256)$ of 400 unique objects. Some examples are shown in the left of Fig. 2. The images contain occluding contours, and for empty background pixels $i, j$ we set $x_0(i, j) = (-1, -1, -1)$. We augment the training data by creating two convex/concave copies of each patch field $x_0^u$ that does not contain any background. At inference time, we use the DDIM sampler [48] with 50 sampling steps and with guidance. Additional details are in the appendix.

For comparison, we consider three deterministic approaches and two stochastic models. SIRFS [4] and Derender3D [53] are deterministic inverse-rendering models that estimate lighting and reflectance

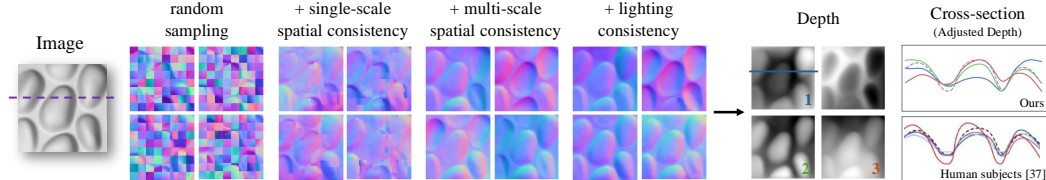

Figure 4: Ablations, and comparison to human subjects using image and psychophysics data from [37]. *Left:* Ablations demonstrate the importance of each component. *Right*: Depth cross-sections extracted from four (integrated) samples from the convex mode of our full model exhibit relief-like variations similar to those reported across human subjects. (The dashed line is the depth that was used to render the input image.)

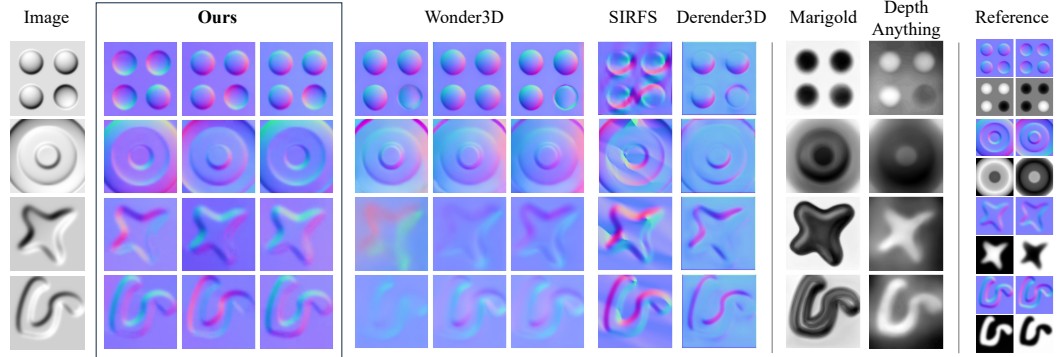

Figure 5: Normals produced by our model for various synthetic test surfaces rendered with directional light sources. For depth maps, brighter is closer. "Reference" depicts the shapes—each with a convex/concave counterpart—that were used to render the input images. We find that our reconstructions are more accurate and diverse than other methods.

together with shape. They are among the few recent models that do not rely critically on having input occluding contour masks. Depth Anything [56] is a recent learning-based deterministic model for monocular depth estimation. It leverages a DINOv2 encoder [39] and a DPT decoder [44] and is trained for depth regression using 62M images. For comparisons to stochastic models, we include Marigold [27] which is derived from Stable Diffusion [45] and is fine-tuned for depth estimation. We also include Wonder3D [33], which likewise leverages a prior based on Stable Diffusion. Wonder3D is trained to generate consistent multi-view normal estimates on more than 30k 3D objects, and it achieves state-of-the-art results on 3D reconstruction benchmarks [12].

## 4.1 Ablation Studies

Figure 4 analyzes the key components of our model using a crop of a shape and image from the lab of James Todd [37] (the complete image is in Appendix A.5). The left of the figure shows that when each patch is reconstructed independently, the resulting normals are inconsistent, because each patch may choose a different concave/convex mode as well as its various flattenings and tiltings. When spatial consistency guidance is applied at one scale, the global field is more consistent but suffers from discontinuous seams due to poor local minima. With multiscale sampling the seams improve, but separate bump/dent regions can still choose different modes without being consistent with any single dominant light direction. Finally, when lighting consistency is added, the output fields become more concentrated around two global modes—one that is globally convex (lit primarily from below) and another that is globally concave (lit primarily from above).

In the right of the figure, we compare samples from our model to depth profiles that were labeled by humans on the same image [37]. (These results have appeared previously in [31].) We generate four samples from our model's globally-convex mode and integrate them to depth maps using [15]. Their cross-sections exhibit variations with similar qualitative structure.

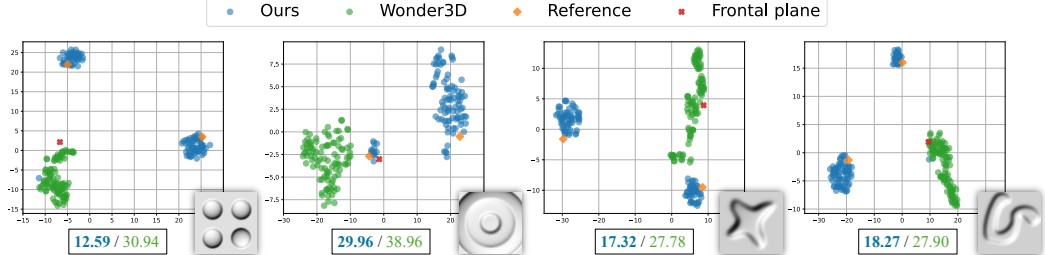

Figure 6: t-SNE visualizations of normal field samples produced by our model and by Wonder3D. Plots depict 100 samples from each model, along with the two mathematical possibilities (under directional light) and the normals of a trivial frontal plane. For each model we report the Wasserstein distance (smaller is better) between its samples and the reference distribution, which is uniform over two possibilities. Our model is more accurate and in all cases covers both possibilities.

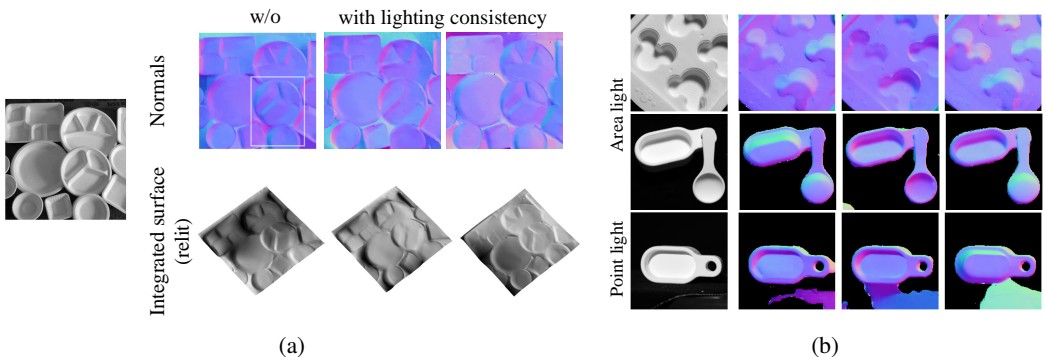

(a)                                                              (b)

Figure 7: Sampled reconstructions for real images. (a) For the 'plates' image from [57], regions such as the indicated box can exhibit independent convex/concave flips when lighting consistency is not used; but when lighting consistency is enforced, only two global modes emerge. (b) Sampled reconstructions for some multistable images we captured with illumination from a point or area light. (Rotate them by $180°$ to enhance the alternative experience.) Note that half of the object in the first row was painted matte, and its other half was left glossy. Despite being trained entirely on synthetic data under idealized lighting, the model exhibits some generalization by producing plausible multistable outputs for these captured scenes.

## 4.2 Ambiguous Images

Figure 5 shows results from our full model for images and shapes that we intentionally design to be ambiguous, using insight from [31]. Each one can be perceived as either convex or concave, as shown in the right-most column (Reference). Samples from our model clearly demonstrate the effectiveness of our model in terms of both coverage and accuracy of the possible shapes. In contrast, we find that the two models derived from Stable Diffusion (Wonder3D and Marigold) provide less accuracy on these images, and that, on average, they tend to have a 'lighting from above' prior baked in. For instance, they tend to interpret the third and fourth row as concave, while Depth Anything [56] interprets them as convex. Additional results are included in the appendix.

Figure 6 visualizes distributions of shape reconstructions as 2D t-SNE plots (with perplexity equal to 30) by sampling 100 random seeds for our model and for Wonder3D. For reference, we also plot the t-SNE embeddings of a frontal plane, $\hat{x}_0(i, j) = (0, 0, 1)$ and of the two reference shapes. Our model covers both reference shapes whereas Wonder3D either covers only one or is close to a plane. These differences in coverage and accuracy are also apparent in terms of Wasserstein distance.

## 4.3 Real Images

We evaluate our model using a few different categories of captured images. In each case, we resize the image to $256 \times 256$ (to accommodate the restrictions of some of the previous models), and we use the multiscale schedule described in Appendix A.10.

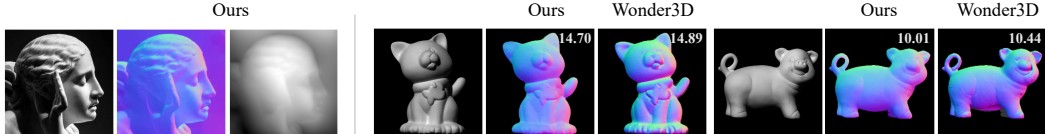

Figure 8: *Left*: Reconstructed normals and integrated depth for an image taken from the web. *Right*: Reconstructions for images in the SfS dataset of [54] with median angular error from the ground truth normals (lower is better). Our model's accuracy is on par with the best existing methods. Additional quantitative results are in the appendix.

**Captured ambiguous images.** Inspired by the 'plates' image in [58], we captured a handful of images that induce multistable perceptions for human observers. We captured these images with a Sony $\alpha$7SIII camera under lighting from area or point light sources, and Fig. 7b shows some examples. We find that our model's multimodality is qualitatively well aligned with perceptual multistability.

**Shape from shading dataset [54].** This dataset contains diffuse objects captured with directional light sources and a dark background, along with ground truth surface normals for measuring accuracy. The right of Fig. 8 shows that our model produces normal maps that are on par with previous methods, even without knowledge of light source directions. Additional quantitative results are in Appendix A.8

**Internet and astronomical images.** The left of Figure 8 shows that our model can produce a detailed and plausible shape estimate for a tone-mapped sRGB image taken from the web [2]. Appendix A.6 includes two satellite images of the surface of Mars and shows that our model reproduces the so-called crater illusion.

## 5   Related Work

Given the recent success of diffusion models in generating realistic images [22, 49, 28], many works have explored the power of patch-based diffusion, including for generating high-resolution panoramic images [3, 32]. Our method also leverages patch diffusion, but it departs from these work in two key ways. Unlike [3, 32], we do not generate patches auto-regressively or require an anchor patch. Instead, we simultaneously guide all patches (e.g., 100 patches for a $160 \times 160$ image) toward a coherent output using spatial consistency guidance. A second distinction is that we do not provide a global condition such as text to each individual patch. Instead, each patch is conditioned only on the corresponding crop of the input grayscale image, which is why we call it a bottom-up architecture. It shares the same spirit as previous work on inverse lighting [14], which also uses a bottom-up architecture to produce a variety of explanations that can then be integrated with top-down information.

Recently, Wang et al. [52] introduced a patch-based diffusion training framework that incorporates patch coordinates to reduce training time and storage cost. Patch-based diffusion has also been used for other tasks. Ozdenizci et al. [42] use overlapping patch diffusion to restore images in adverse weather, and Ding et al. [11] use it to synthesize images in higher resolution. All of these works use fairly large patches (e.g., $64 \times 64$) and some of their components, such as feature-averaging or noise-averaging, are not appropriate for our shape from shading problem because of its inherent multi-modality. A convex sample and a concave sample cannot simply be averaged to improve the output. These challenges motivate the novel features of our model, including global consistency guidance and multiscale sampling.

## 6   Conclusion

Inspired by the multistable perception of ambiguous images, and by mathematical ambiguities in shape from shading, we introduce a diffusion-based, bottom-up model for stochastic shape inference. It learns exclusively from observations of everyday objects, and then it produces perceptually-aligned multimodal shape distributions for images that are different from its training set and that appear ambiguous to human observers. A critical component of our model is a sampling scheme that

operates across multiple scales. Our model also provides compositional control: global lighting consistency can be turned on or off, thereby controlling whether regional bumps/dents can each undergo concave/convex flips independently. Our findings motivate the exploration of other multiscale stochastic architectures, for a variety of computer vision tasks. They may also help improve the understanding and modeling of human shape perception.

**Limitations**    A key limitation is our restriction to textureless and shadowless Lambertian shading. While this restriction is common in theoretical work [29, 54, 20] and useful for creating ambiguous images, it is well-known that many ambiguities disappear in the presence of other cues such as glossy highlights, cast shadows, and repetitive texture. Also, since our model is predominantly bottom-up, it suffers when large regions of an image are covered by cast shadows (e.g., the shoulder region in Fig. 8). These types of regions often require non-local context like object recognition in order to be accurately completed. Incorporating more diverse materials (e.g., as in [14]) and top-down signals into our model are important directions for future research.

Another limitation of our model stems from its sequential V-cycle approach to multiscale sampling. It scales linearly with the number of resolutions, which is likely to be improved by optimization or parallelization that increases runtime efficiency. Also, since our multiscale approach is training free, it requires a manual search to identify a good schedule. Similar to previous work that restarts the sampling process from intermediate timesteps [55], ours also require choosing the timestep at which to resume sampling. Overall, further analysis is needed to better understand the structure of our model's latent space, and to discover more efficient and general approaches to multiscale generation.

### Acknowledgments

We thank Jianbo Shi for helpful discussions and Steven Zucker for suggesting the crater illusion. We also thank James Todd, Benjamin Kunsberg, Steven Zucker and William Smith for kindly sharing their images and perceptual stimuli. This work was in part supported by JSPS 20H05951, 21H04893, and JST JPMJAP2305. It was also in part supported by the NSF cooperative agreement PHY-2019786 (an NSF AI Institute, http://iaifi.org).

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

# A   Appendix / supplemental material

## A.1   Algorithms pseudocode

We provide the pseudocode for the single-scale spatial consistency guided sampling (Alg. 1) and the lighting consistency guidance (Alg. 2). Here, we have hyperparameters $\lambda$ for weighting the smoothness and integrability loss, $\eta_t$ as guidance update weight and $J_t$ as the number of noise update steps. The results in our paper use $\lambda = 0.5$ and $J_t = 3$. The parameter $\eta_t$ is resolution-dependent and is included with the schedule specification in Appendix A.10.

## A.2   Experimental setup details

**Model Architecture.**   We use a conditional UNet architecture similar to [22] with input spatial dimensions of $16 \times 16$ and 4 channels. The input to the UNet is a concatenation of the grayscale shading image and a 3-channel normal map. We use a linear attention module [46] for better time and memory efficiency. The UNet consists of 4 downsampling and upsampling stages composed of the commonly used ResNet [19] blocks, group normalization layers, attention layers, and residual connections.

**Dataset and Training Details.**   We train the pixel-space conditional diffusion model on a dataset that we build from the UniPS dataset [26]. It contains about 8000 $256 \times 256$ synthetic images of 400 unique objects from the Adobe3D Assets [1] rendered from different viewing directions. We render the objects with the shadow-less Lambertian model using the provided ground truth normal maps in [26], a randomly sampled directional light source within 60 degrees of the $z$-axis, and an albedo value in $[0.5, 1]$. The surface normal values outside of the objects are set to $(-1, -1, -1)$.

We subdivide the image into non-overlapping patches of size $16 \times 16$ and train our model to predict the noise at each sampled timestep using a smooth L1 loss. To train the diffusion UNet, we use the cosine variance schedule [38] with 300 timesteps. The model is trained using the AdamW optimizer for 500 epochs with learning rate 2e-4. It takes about 40 hours using one Nvidia A100 GPU.

---

**Algorithm 1:** Spatial Consistency Guided Sampling at a Single Scale

---

**Data:** $\{x_t^u, c^u\}_{u \in \mathcal{V}}; \epsilon_\theta, \{\eta_t, \lambda\}$

1  **while** $t > 0$ ;                                                       // Parallel guidance
   **do**
2  $\quad$ **for** $j = 1, 2, \cdots, J_t$ ;                     // Gradient descent for multiple steps
   $\quad$ **do**
3  $\quad\quad$ $\hat{x}_0^u = \text{Pred}(x_t^u; c^u, \epsilon_\theta)$ ;                            // Predict $\hat{x}_0^u$ using Eq.4
4  $\quad\quad$ $\hat{x}_0 = \text{Reshape}(\{\hat{x}_0^u\})$ ;                          // Patch to global layout
5  $\quad\quad$ $\mathcal{L}(\hat{x}_0) = \frac{1}{|\mathcal{E}|}\mathcal{L}_S(\hat{x}_0^u, \hat{x}_0^v) + \lambda\frac{1}{|\mathcal{V}|}\mathcal{L}_I(\hat{x}_0^v)$ ;                            // Eq. 8
6  $\quad\quad$ $x_t^u \leftarrow x_t^u - \eta_t \cdot \nabla_{x_t^u}\mathcal{L}(\hat{x}_0)$
   $\quad$ **end**
7  $\quad$ $x_{t-1}^u \leftarrow \text{Denoise}(x_t^u; c^u, \epsilon_\theta)$ ;     // Parallel denoising (Eq.3) for each patch
   **end**
   **return** $\{x_0^u\}_{u \in \mathcal{V}}$ ;                            // Denoised normal prediction

---

**Algorithm 2:** Lighting Consistency Guidance

---

**Data:** $\{x_0^u, c^u\}_{u \in \mathcal{V}}$

1  $\hat{l}^u = \text{Robust Infer}(x_0^u, c^u)$ ;            // Infer light source direction as in Eq. 11
2  Cluster center $\{L_1, L_2\}$, assignment $k^u$ = K-Means Clustering($\{\hat{l}^u\}$, # clusters = 2)
3  **for** $u \in \mathcal{V}, k^u = 2$ ;                                      // Assume that $|k^u = 1| \geq |k^u = 2|$
   **do**
4  $\quad$ $(x_0^u)_\mathbf{x} \leftarrow -(x_0^u)_\mathbf{x}, (x_0^u)_\mathbf{y} \leftarrow -(x_0^u)_\mathbf{y}$;     // Convex/concave flip of the normals
   **end**
   **return** $\{x_0^u\}_{u \in \mathcal{V}}$

---

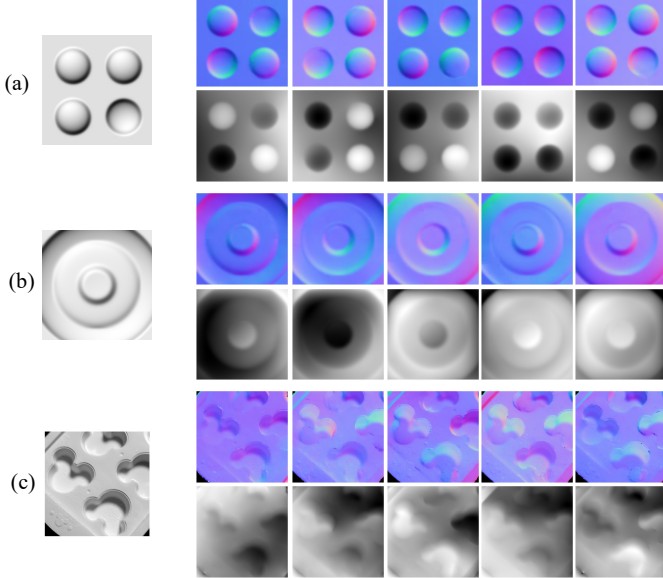

Figure 9: Sampled output normals and integrated depth maps when lighting consistency is not enforced. In contrast to the samples in Fig. 5, regions within the same image can undergo independent convex/concave flips.

**Data Augmentation.** During training, we augment the dataset with convex/concave flips of interior patches, meaning those that do not include any occluding contour and background. In Sec. A.7, we compare two models trained with and without such augmentation. While the augmentation leads to a more balanced multimodal output distribution and thus smaller Wasserstein distance, our model trained without augmentation is already capable of producing multistable outputs on test images.

**Sampling.** We use the DDIM sampler [48] with 50 sampling steps with guidance. Details of the multiscale sampling schedule and guidance learning rate $\eta_t$ can be found in Appendix A.10.

**Visualization and Evaluation Metric.** We visualize the output normal map from multiple samples to show the multistable reconstructions. We also show depth maps by integrating the normals using the method by Frankot & Chellappa [15].

To produce the t-SNE visualizations in Figs 6 and 13, we draw 100 samples from each model, and then downsize the sampled normal maps to $64 \times 64$ resolution for computational efficiency. Then we project the samples to a 2-dimensional space using t-SNE with perplexity value of 30. To compute the 1-Wasserstein distance, we set the ground truth distribution to be a uniform distribution over the two reference normal maps: the one used to render the input image, and its convex/concave flip. When computing the 1-Wasserstein distance of model outputs to the ground truth distribution, all normal maps are first downsampled to $64 \times 64$ resolution.

**Baseline models.** For testing with Wonder3D [33], we extract the frontal view prediction and use the default setting in their online demo and code base with a crop size of 256 by 256, classifier-free guidance scale of 3, and 50 sampling steps.

### A.3 Ablation study on lighting consistency guidance

In Fig. 9, we show additional samples from our model where lighting consistency guidance is turned off. Results show that in this case the output distribution of our model allows different regions within a surface (e.g., each dimple, ring or mouse-shape) to undergo an independent convex/concave flip. This is in contrast to Fig. 5 of the main paper, where we see only two global modes.

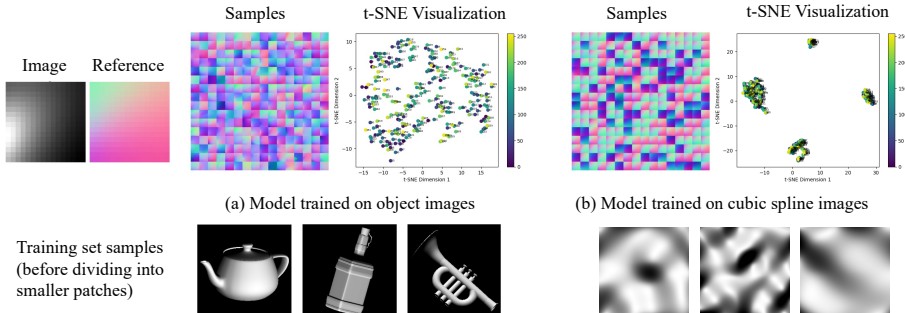

Figure 10: Output normal maps and their t-SNE visualizations when our model is applied to a $16 \times 16$ image of an exactly quadratic surface under directional lighting. When our model is trained using images of spline surfaces (b), the outputs cluster around the four mathematical interpretations from [54] (convex, concave, and two saddles). When it is trained using images of everyday objects (a), the outputs exhibit more diversity. In both scenarios, samples are drawn independently without guidance.

## A.4    Relation to the four-way convex/concave/saddle ambiguity

Figure 10 examines the relationship between our model and the mathematical results from Xiong et al. [54], which show that, in general, an image of an exactly quadratic surface under unknown directional lighting can be explained by four quadratic shapes (convex, concave, and two saddles). If our model is consistent with the theory, we would expect its output shape distributions for such images to be concentrated around four distinct modes that match the four distinct possibilities. When we apply our model to images of exactly-quadratic surfaces like the one in the left of Fig. 10, we find that its output distribution is *not* concentrated near four modes (middle panel in the figure). However, we find that the four-mode behavior emerges when the model is retrained on a different dataset comprising random cubic splines (right panel), which by construction contain a much higher proportion of exactly-quadratic surface patches.

One potential explanation for this behavior is that exactly-quadratic surface patches are too rare in everyday scenes for a vision system to usefully exploit. This may relate to the perceptual experiments in [51] that suggest humans also struggle to perceive the four distinct shapes for such images.

## A.5    Additional results on perceptual stimuli

Figure 11 shows our model's output for images that were used to study human perception in [31] and [37]. Our model produces plausible shapes and multimodal output distributions, while other models sometimes fail to recover a plausible shape or produce only one of the global concave/convex possibilities. The bottom row is an image of several bumps ('cobbles') lit from below. We observe that Wonder3D [33], Marigold [27] and SIRFS [4] interpret the bumps as concave, while Depth Anything [56] interprets them as convex.

## A.6    Additional results on astronomical images; relation to the crater illusion

Figure 12 shows results for two satellite images of the surface of Mars. In images like these, humans often misperceive craters as mountains and vice versa, perhaps due to their bias toward lighting from above. (This is sometimes called the crater illusion.) We tested our model, the diffusion-based models [33, 27], and Depth Anything [56]. Our model sees both the crater and mountain possibilities, but the other models only see one of the two.

Data acquisition is expensive in these situations, so there could be benefit to having a monocular vision model that can automatically produce the multitude of explanations, thereby allowing all possibilities to be examined by gathering additional context. Similar to human perception, in astronomical imaging it is beneficial to have access to all of the possible "bottom-up" explanations, so that one can use context or "top-down" information as effectively as possible.

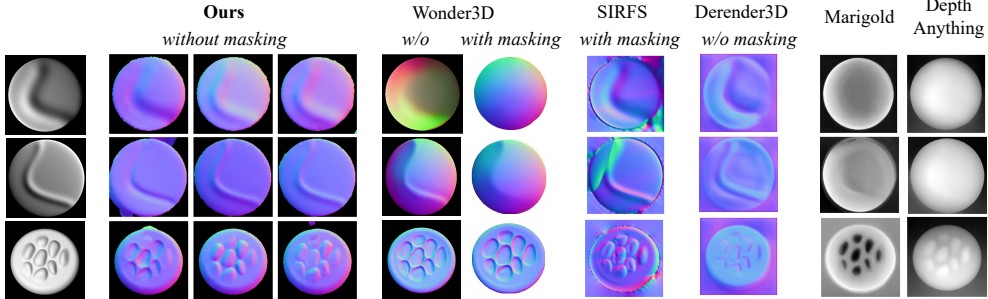

Figure 11: Inferred normal map samples from our model on ridge images taken from [31] and 'cobble' test image from [37].

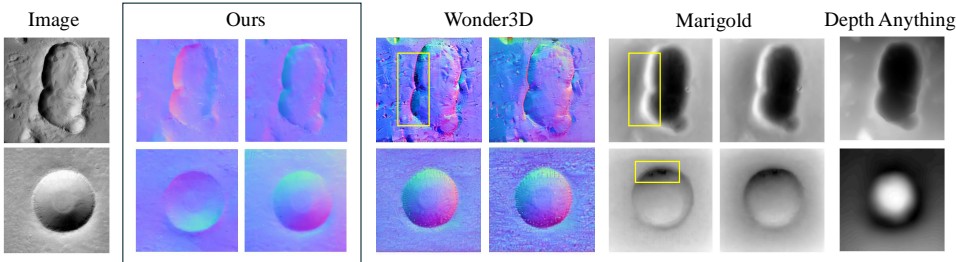

Figure 12: Shape from Martian crater images. For diffusion-based models we show two samples that qualitatively represent all of the 25 samples that we generated for that model. For depth models (Marigold and Depth Anything) brighter is closer. Our model outputs both possibilities, crater and mountain, while other models only see one of the two. Outputs from Wonder3D and Marigold also include artefacts (yellow boxes), with sharp spurious variations in normals or depth.

The two images are taken from:

(1) A triple crater in Elysium Planitia on Mars. Credit: NASA/JPL/University of Arizona. `https://www.universetoday.com/118581/amazing-impact-crater-where-a-triple-asteroid-smashed-into-mars/`

(2) A fresh impact crater, about 3 kilometers wide, gouged from a lava-covered plain in the Lunae Planum region of Mars. `https://skyandtelescope.org/astronomy-resources/astronomy-questions-answers/is-it-possible-that-photos-of-lunar-or-martian-landscapes-show-craters-as-blisters/`

## A.7 Ablation of dataset augmentation

To explore the effect of convex/concave data augmentation during training (see the top of Section 4), we perform an ablation in which we train a model from the same set of patches but without the augmentation. Since the original patches from our training set tend to be dominated by convexity, we expect this to have an affect on the model's response to ambiguous images. Figure 13 and Table 1 bear this out. The figure and table show the same t-SNE visualizations and Wasserstein distances as in the main paper, but this time with without augmentation during training. The model without augmentation exhibits some multistability, especially for the last two images, but it tends to provide less diversity, especially for circular shapes. We hypothesize that this is caused by the existence of predominantly convex spherical shapes in the training set.

## A.8 Quantitative results on a shape from shading benchmark

Table 2 shows quantitative comparisons using photographs from the dataset in [54]. We follow prior practice and report the median angular error of the predicted normal field, where the angular error at each pixel $i$ is $AE(\hat{n}_i, \hat{n}_i^{gt}) = \left| \cos^{-1}(\hat{n}_i \cdot \hat{n}_i^{gt}) \right|$. Diffusion-based outputs are stochastic and

Table 1: Wasserstein distance on multistable perception stimuli

| Model | (a) four circles | (b) nested rings | (c) star | (d) snake |
|---|---|---|---|---|
| Wonder3D[33] | 30.94 | 38.96 | 27.78 | 27.90 |
| Ours (w/o data augmentation) | 33.79 | 39.56 | 22.04 | 23.02 |
| Ours | **12.59** | **29.96** | **17.32** | **18.27** |

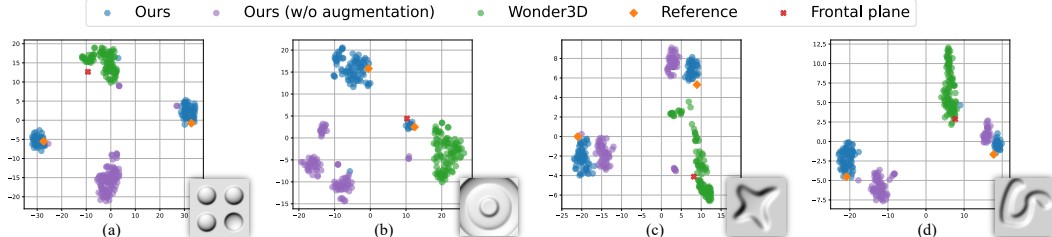

Figure 13: t-SNE plots showing that our model trained without flip augmentation already exhibits multistable outputs, especially on the last two images. Training with data augmentation improves the overall diversity on those test images.

potentially multi-modal (e.g., with global convex or concave possibilities), so for these we report the average error of the top-five predictions taken from 50 independent runs. Note that the method from [54] requires the true light direction to be provided as input, whereas ours and others do not.

Overall, our model shows competitive performance on this benchmark compared with existing methods. We notice that our model trained without data augmentation via per-patch concave/convex flips (see the top of Section 4) performs better, while our model trained with data augmentation has slightly worse performance, likely due to its larger per-patch search space. This may be related to other quality-versus-diversity trade-offs that have been observed in previous conditional diffusion models [10, 21]. We leave for future research questions of how to achieve better combinations of quality and diversity, and how to incorporate other cues such as occluding contours and top-down recognition cues.

### A.9 Ablation on lighting distribution in training set

Figure 14 shows an experiment where we change the distribution of light source directions in the training set. The two models A and B are trained on the same shapes without any data augmentation, but Model B has 80% of the images lit from above. Model A is trained with uniformly sampled lighting from above and from below. We test both models on an image that appears concave when 'lit from above'. From 50 random samples and their t-SNE projections, we see that Model B's distribution is biased towards the concave answer while Model A shows a more balanced distribution. This shows that lighting bias in the training set can have an effect on the output distribution.

Table 2: Shape from shading benchmark quantitative results. Errors are measured by median angular error of normals maps. Model performance for diffusion based models is averaged over the top 5 estimates over 50 random seeds.

| Model | cat | frog | hippo | lizard | pig | turtle | scholar |
|---|---|---|---|---|---|---|---|
| Xiong et. al [54] (known lighting) | 14.83 | **11.80** | 20.25 | **12.70** | 15.29 | 17.90 | 28.13 |
| SIRFS Cross-Scale [4] | 20.02 | 19.86 | 21.00 | 23.26 | 13.17 | 11.96 | 25.80 |
| Wonder3D [33] | 14.29 | 18.20 | 16.81 | 15.70 | 10.10 | 9.59 | 25.32 |
| Ours (w/o data augmentation) | **11.49** | 15.56 | **14.17** | 13.30 | **9.27** | **8.72** | **22.01** |
| Ours | 14.95 | 21.46 | 15.94 | 12.82 | 11.98 | 9.99 | 27.21 |

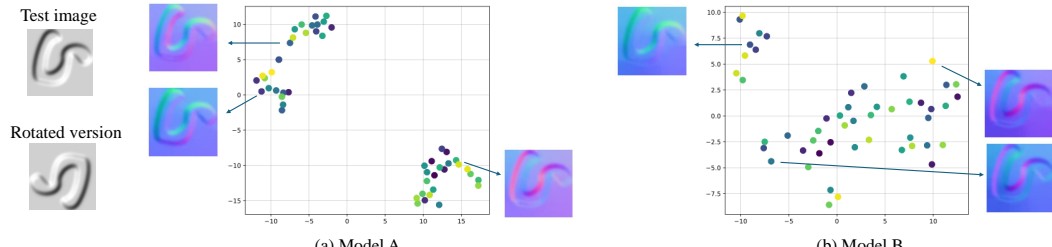

(a) Model A              (b) Model B

Figure 14: Effect of lighting bias during training. Model A is trained using synthetic images with lighting directions that are uniformly sampled within a $60°$ cone around the view direction. Model B is trained using the same shapes but with lighting that is distributed non-uniformly within the cone, where 80% of images are lit from above. After training, we draw 50 samples from each model for a test image, using the same schedule and guidance hyperparameters. We project the results using t-SNE (dots are randomly colored for visual clarity) and show representative samples. Model A produces a balanced distribution across convex and concave explanations, whereas Model B produces concave predictions more often. (To humans, the test image usually appears concave, and it usually appears convex when rotated. These are both physically consistent with lighting from above.)

Table 3: Multiscale optimization schedule

|  | Perception Stimuli | Captured Photo |
|---|---|---|
| Resolution | [160, 128, 64, 80, 96, 112, 128, 144, 160] | [256, 160, 96, 128, 192, 224, 240, 256] |
| Guidance rate $\eta$ | [20, 15, 10, 10, 10, 15, 15, 20, 20] | [30, 20, 12, 15, 20, 25, 28, 30] |
| Lighting guidance | [T] $\times$ 2 + [F] $\times$ 7 | [F] $\times$ 3 + [T] $\times$ 2 + [F] $\times$ 3 |
| $N\&R$ start $t$ | [300] + [232] $\times$ 8 | [300] + [238] $\times$ 7 |
| Runtime (seconds) | 105s (single Quadro RTX 8000) | 125s (single Quadro RTX 8000) |

## A.10 Multiscale schedule specification

In Table 3 lists the scheduler hyperparameters for multiscale guided sampling. We use a notation for lists with repeating elements, where concatenation is represented as follows: for example, $[A] \times 2 + [B] \times 3$ denotes the list $[A, A, B, B, B]$.

When designing the multiscale schedule for inference, we find it helpful to have consecutive resolutions that are not integer multiples of the previous one, especially in the coarse to fine direction. This leads to improved quality because pixels that are adjacent to patch seams at one resolution become interior to a patch at another resolution.

For the initial resolution (our experiments use $160 \times 160$ or $256 \times 256$), guidance is applied only after the 8th DDIM denoising step since the predicted $\hat{x}_0$ at very early stages are often not informative enough for guidance. In terms of lighting consistency guidance, we find that it is often not necessary to apply at every resolution for a perceptually consistent normal estimation.

We apply normal field fusion using the last three resolutions, after the spatial predictions and choices of per-patch convex/concave modes have stabilized.

