# OpenReview forum: "Multistable Shape from Shading Emerges from Patch Diffusion"
_NeurIPS.cc/2024/Conference — NeurIPS 2024 spotlight_

### Official Review · Reviewer_E5e1 · 2024-07-12

**Soundness:** 3
**Presentation:** 3
**Contribution:** 2
**Rating:** 5
**Confidence:** 4

**Summary:**

This paper deals with the problem of normal map estimation from images. The authors trained a conditional diffusion model to sample a normal map output given an image input by processing 16x16 patches. The paper suggests a multiscale approach for resampling consistently at multiple scales. Due to the model’s non-deterministic property, it can cover possible outputs in ambiguous cases. The model is also integrated with a lighting consistency module, to verify that each possible output normal map is consistent with a global directional lighting.  Experiments show generalization to unseen cases, for both synthetic and real images.

**Strengths:**

* A novel diffusion model for consistently predicting normal maps from an input image
* Clear presentation of the model.
* Dealing with ambiguous cases consistently.
* Efficient and lightweight model.
* Strong generalization to unseen images.

**Weaknesses:**

Limited applicability. Most of the tested images are either synthetic or captured in a controlled setup. The compared baseline methods are not limited to textureless and shadowless Lambertian shading.

**Questions:**

* Are there real-world applications that could be solved or improved by the suggested method?
* Could this method be used in multiview settings, for consistently reconstructing texture-less surfaces?

**Limitations:**

The authors discussed limitations and societal impact.

---

> ### Author Rebuttal · Authors · 2024-08-06
>
> **Weaknesses**:
> > Limited applicability. Most of the tested images are either synthetic or captured in a controlled setup. The compared baseline methods are not limited to textureless and shadowless Lambertian shading.
>
> Yes, it is absolutely true that the comparison methods are not limited to Lambertian surfaces, but it is also true that they cannot produce multistable solutions, as we demonstrate.
>
> We chose to focus on Lambertian surfaces because there are rich theoretical and perceptual characterizations of the ambiguities, as described in Section 2.1. Focusing on Lambertian surfaces allows us to compare our model’s outputs to those predicted by theory, and in Figures 5 and 6, we indeed see that our model produces both discrete modes and continuous GBR families.
>
> We will explore expanding our model to non-Lambertian (e.g., glossy) surfaces with global light transport, including cast shadows and interreflections. These phenomena produce additional visual cues that dramatically reduce the ambiguity in many cases, but since they are not always present, one cannot rely on them entirely. It is an interesting open challenge to create models that can properly exploit these cues when they are present, and that can also avoid overfitting by correctly producing diverse local and global shapes when they are absent.
>
> **Questions**:
> > Are there real-world applications that could be solved or improved by the suggested method?
>
> Please see the global response.
>
> > Could this method be used in multiview settings, for consistently reconstructing texture-less surfaces?
>
> Thank you for the great suggestion. Since correspondences are harder to detect for texture-less surfaces (see e.g., [%]), shape-from-shading models like ours can be a useful complement. To adapt our model to a multiview setting, one could add additional terms to the guidance loss or consistency module, verifying agreement between views. We will add this point to the conclusion section.
>
> * [%] Cryer JE, Tsai PS, Shah M. Integration of shape from shading and stereo. Pattern recognition. 1995 Jul 1;28(7):1033-43.

---

> > ### Comment · Reviewer_E5e1 · 2024-08-11
> >
> > I appreciate the author's response. To me, the method still feels technical without contributing any deep insights into the nature of the problem. The method is presented as comparable to Wonder3D, with the extra ability to sample multiple valid solutions, but in terms of applicability, Wonder3D is much more practical and can work on various real-world images. In contrast, the presented method was only tested on a very synthetic setup. While the architecture is novel and the results are nice, without either deeper insights or more practical contributions, I remain slightly positive about this paper but stay with my original recommendation for now.

---

> > > ### Author Response · Authors · 2024-08-14
> > > **Further response to Reviewer E5e1**
> > >
> > > Thank you for your feedback. In case it helps, here are some of the insights that we gained while working on this paper:
> > >
> > > **1. When done right, a model can reliably mimic multistable shape perception.** Before our paper, there was no computational model that was consistent with the human phenomena of GBR-like ambiguities and multistability. For decades, computational research has largely ignored this inconsistency, while perception researchers have bemoaned it (e.g., [47]). Our paper finally shows it is possible to bridge this gap, and moreover, that we can do it using a small 10MB model and only 400 training shapes. If someone had told us two years ago that this was possible—and that they could so easily generalize from synthetic data to captured photographs—we would have been very skeptical.
> > >
> > > **2. There is a practical way to model the “right kind” of lighting consistency.** For decades, lighting has been an “elephant in the room” for this problem. Researchers in both machine and human vision have long acknowledged that ambient occlusion, interreflections, etc. are substantial and unavoidable, but they have had few ideas for how to create computational models that can succeed in spite of them. (A notable attempt is D.A Forsyth, “Variable-Source Shading Analysis”, IJCV 2011.)
> > >
> > > Our lighting constraint in Section 3.1 is a surprisingly simple way to enforce consistency that is “just right”. It is not too strong to be intolerant to deviations from idealized lighting and reflectance, yet it is strong enough to achieve globally-consistent explanations.
> > >
> > > Moreover, by being selectively enforceable (e.g., our Appendix Figure 9), it finally opens an avenue for research into where and when humans choose to apply such consistency. (If you have not already done so, it’s worth seeing Figure 4 in (Ostrovsky et al., 2005), which is Ref. [*] in our response to Reviewer xHhf. Their digitally altered photographs demonstrate quite clearly that humans apply global lighting consistency in some places but not in others.)
> > >
> > > **3. Compositionality is valuable.** In hindsight this should not be a surprise because compositionality has long been recognized as important for efficient learning and generalization. But the effectiveness of a simple composition of the same patch model over space and scale is something that surprised us in this project.
> > >
> > > **4. There may be a probabilistic explanation for the conundrum of the four-way choice.** For at least 15 years (see [48]), it has been known that four shapes (convex, concave and two saddles) are mathematically consistent with an image of an exactly-quadratic surface, but that humans almost always perceive only two of them (convex, concave). In Appendix Section A.4., our model provides a way to explain this probabilistically: Exactly-quadratic surface patches may simply be too rare in everyday scenes for the human visual system to usefully exploit them.

---

### Official Review · Reviewer_3BEx · 2024-07-13

**Soundness:** 4
**Presentation:** 4
**Contribution:** 4
**Rating:** 8
**Confidence:** 4

**Summary:**

The authors argue that monocular shape reconstruction models ought to produce distributions of outputs rather than point estimates or tight distributions, to adequately cope with known ambiguities, and draw additional motivation from multistable perception in humans. By training a small patch-based diffusion model, the authors convincingly demonstrate multistable solutions to shape from shading through a fairly lightweight algorithm.

**Strengths:**

- Proposes multistable perception to cope with the ambiguity of monocular shape reconstruction.
- Presents a viable solution based on a small patch-based diffusion model and a fairly simple consistency-enforcing algorithm.
- Demonstrates compelling results for coping with ambiguities, even when the model was trained on synthetic renderings of common 3D objects.
- Very well written and easy to read.

**Weaknesses:**

Nothing major, other than the contribution being more conceptual in nature, with no immediate practical impact.

**Questions:**

Technical:
========
- L41: I wonder if training on common objects free of illusions, similar to what humans are most used to, is necessary for the resulting models to not be biased to any particular interpretation. That is, in comparison to pre-trained diffusion models who seem to be biased to "lighting from above." Since the diffusion model employed is small, it would be nice to include such an experiment and assess the level of bias in the generated reconstructions in terms of related biases in the training data.
- L208: why the 3D objects from [23]?
- L207: why the 0.7 lower bound?
- L31: it would be nice to go back and relate the proposed algorithm to the principle of least commitment.

Presentation:
==========
Section 2
- L103: "learned" Gaussian transitions
  - The forward process, with a predetermined noise schedule, doesn't seem to entail any learning, right?
- L107: $\mathcal{N}(x_{t-1}, \cdot)$ -> $\mathcal{N}(x_t, \cdot)$
- L118: $\epsilon(x_t; c)$ -> $\epsilon(x_t, t; c)$
Section 3
- L128: *differentiable surface* sounds off. What's a better way to say this? A representation?
- Eq. 6: flattening the $d \times d$ patch is a bit counterintuitive, and was later abandoned for Eq. 8.
- Figure 3 is quite confusing. Could use more work.
Section 5
- L271: We do provide -> We do not provide (?)

Nitpicking:
========
- L38: Instead we approach [..] and *are* (?) inspired
- L134: It would have been nice to be able to refer to the equation on L109.
- L142: Better to say something about $\mathcal{E}$ by way of introduction or definition.

**Limitations:**

As the authors point out, this study inherits some of the restrictions common in theoretical work. In addition, the proposed model while powerful is not particularly fully developed or optimized. That said, none of that seems to diminish the value of the contribution in terms of either the concept or the obtained results demonstrating the capabilities and potential of such reconstruction models.

---

> ### Author Rebuttal · Authors · 2024-08-06
>
> We thank the reviewer for the positive feedback and the detailed suggestions!
>
> **Weaknesses & Limitations**:
>
> Please see the global response.
>
> **Questions**:
>
> **Technical**
> > L41: I wonder if training on common objects free of illusions, similar to what humans are most used to, is necessary for the resulting models to not be biased to any particular interpretation. That is, in comparison to pre-trained diffusion models who seem to be biased to "lighting from above." Since the diffusion model employed is small, it would be nice to include such an experiment and assess the level of bias in the generated reconstructions in terms of related biases in the training data.
>
> This is a great question. We believe that diffusion models like Wonder3D and Marigold show such bias not due to the type of objects in their training sets, but due to the distribution of illumination conditions in their training sets, which are likely biased to being “from above”. In contrast, our synthetic training dataset is rendered with randomly sampled light source directions. In the attached **PDF Fig. R2**, we show an experiment where we change the distribution of light source directions in the training set. The two models A and B are trained on the same shapes without any data augmentation, and the only difference is that for model B we make 80% of the images lit from above. We then test on an image that looks concave when one holds the ‘lighting from above’ prior. From 50 random samples and their t-SNE projections, we see that model B’s distribution is biased towards the concave answer while model A shows a more balanced distribution. This shows that lighting bias in the training set can have an effect on the output distribution.
>
> The bias in existing diffusion models could also come from their fine-tuning schemes, but more studies are required for a thorough answer.
>
> Please also revisit Appendix Fig. 10, which shows another effect of dataset bias. The model in (b) is trained solely on cubic-spline images, and it behaves differently from our main model for certain inputs. When given a quadratic-patch image, it produces four distinct clusters that  correspond to the well-known convex/concave/saddle ambiguity. However, our main model (a), which is trained on everyday objects, produces a qualitatively different distribution.
>
> > L208: why the 3D objects from [23]?
>
> We were inspired by the success of using the dataset of [23] for a different 3D reconstruction task, namely photometric stereo [23, #], and in particular that work’s successful generalization from synthetic to real images. Our results show analogous generalization to unseen images.
>
> * [23] Universal Photometric Stereo Network using Global Contexts. Satoshi Ikehata. CVPR 2022.
>
> * [#] Scalable, Detailed and Mask-free Universal Photometric Stereo. Satoshi Ikehata. CVPR 2023.
>
> > L207: why the 0.7 lower bound?
>
> Thank you for asking. We checked our code again and found our lower bound is actually 0.5 instead of 0.7. We apologize for the error and will update this in the paper.
>
> Training with a randomly sampled albedo instead of a fixed unit albedo helps the model generalize to images that have different exposure levels. We found that training with fixed albedo causes the model to produce overly-flattened shapes in under-exposed images.
>
> We chose the lower bound of 0.5 because white paint often has albedo between 0.5 and 0.9, and because empirically we found it sufficient for generalization to real images.
>
> > L31: it would be nice to go back and relate the proposed algorithm to the principle of least commitment.
>
> Great suggestion. We will add a sentence in the conclusion to connect back to the least commitment principle.
>
> **Presentation**
>
> > L103: "learned" Gaussian transitions. The forward process, with a predetermined noise schedule, doesn't seem to entail any learning, right?
>
> Yes this word should be removed as there is no learning component.
>
> > L107: $\mathcal{N}(x_{t-1}, \cdot)$ -> $\mathcal{N}(x_t, \cdot)$
>
> Here $x_{t-1}$ denotes the random variable and will be the same on both sides of the equation. We have also verified this with the DDPM paper [19] (Ho et al., 2020).
>
> > L118: $\epsilon (x_t; c)$ -> $\epsilon (x_t, t; c)$
>
> Yes, thank you very much.
>
> > L128: differentiable surface sounds off. What's a better way to say this? A representation?
>
> Yes. We will change it to *a surface represented by a differentiable height function $h(x, y)$*.
>
> > Eq. 6: flattening the $d \times d$ patch is a bit counterintuitive, and was later abandoned for Eq. 8.
>
> Agreed. We will change the indexing in Eq. (6) to (i, j)-notation, making it consistent with Eq. (8).
>
> > Figure 3 is quite confusing. Could use more work.
>
> We agree that it can be improved. We have proposed a revised version in the **PDF Fig. R3** and we welcome feedback.
>
> > L271: We do provide -> We do not provide (?)
>
> Yes. We do NOT provide global conditioning here.
>
> > L38: Instead we approach [..] and are (?) inspired
>
> Thank you for the suggestion. We will make the sentence clearer.
>
> > L134: It would have been nice to be able to refer to the equation on L109
>
> Yes. We will improve the paragraph and add the equation label.
>
> > L142: Better to say something about $\mathcal{E}$ by way of introduction or definition.
>
> Great advice. We will add descriptions of the edge set as well.

---

> > ### Comment · Reviewer_3BEx · 2024-08-11
> > **Thank you**
> >
> > I'm satisfied with the answers to my comments, and encourage the authors to reflect the technical clarifications in the main paper. It would also help to include more specific references to the different appendices, prompting the readers more explicitly to those additional experiments.

---

### Official Review · Reviewer_xHhf · 2024-07-15

**Soundness:** 2
**Presentation:** 2
**Contribution:** 2
**Rating:** 5
**Confidence:** 3

**Summary:**

This paper presents a patch-wise diffusion based Shape-from-shading strategy to recover multiple shapes satisfying concave/convex ambiguity from the images. It models shape inference as a generative process conditioned by light intensities. The generative process is governed by  small, non-overlapping patches at multiple scales. Simple measures of overall spatial consistency and global light consistency allows the disjoint patches to converge to globally coherent shapes.

**Strengths:**

1. The use of patch-wise diffusion is an  interesting for the proposed methodology. By breaking the images into small patches, allows the method to generalise beyond the shapes belonging to the training datasets. It also allows to preserve the concave/convex ambiguities.

2.  Experiments show that the method is able to reconstruct shapes unto convex/concave ambiguities well, even on the datasets on which it was not trained on.

3. Ablation studies are a plus.

**Weaknesses:**

The reconstruction (figure 8) obtained by the proposed method is quite accurate, quite similar to Wonder3D. However, since the method yields multiple solutions, how is this particular solution chosen and what is the range of variations in the different solutions obtained.

**Questions:**

Since the method yields multiple solutions satisfying concave/convex ambiguities. I am wondering if it is possible to determine the exact number of solutions. For example, in figure 5, 3 solutions are reported. However, it is easy to see that there are only 2 solutions possible in these cases satisfying the discrete (convex/concave ambiguity are possible.  The third solution is normally a bas-relief of one of these two solution.  Therefore, it would be nice to add a mechanism to classify the solutions according to possible ambiguities.

**Limitations:**

yes

---

> ### Author Rebuttal · Authors · 2024-08-06
>
> **Weaknesses**:
> > The reconstruction (figure 8) obtained by the proposed method is quite accurate, quite similar to Wonder3D. However, since the method yields multiple solutions, how is this particular solution chosen and what is the range of variations in the different solutions obtained.
>
> The samples shown in Fig. 8 are drawn from the 10 most accurate reconstructions from among 50 samples, for both our model and Wonder3D. Since both models are stochastic, the median angular error can vary from sample to sample. Figure 8 conveys how close each method comes to having the ground truth shape within its output distribution.
>
> The following chart shows the diversity of the top 10 samples for each method, measured by standard deviation of error (in degrees) with respect to ground truth.
>
> |  | cat | frog | hippo | lizard | pig | turtle | scholar |
> |----------|----------|----------|----------|----------|----------|----------|----------|
> |   Wonder3D  |   $14.42\pm0.17$  |  $18.46\pm0.33$  |   $16.97\pm0.18$  |   $15.81\pm0.12$  |  $10.22\pm0.15$  |   $9.69\pm0.12$  |   $25.55\pm0.31$  |
> |   Ours  |   $15.49\pm0.68$  |   $22.41\pm1.13$  |   $16.59\pm0.82$  |   $13.66\pm1.03$  |   $13.03\pm1.15$  |   $10.53\pm0.63$  |   $27.95\pm0.93$  |
>
> In terms of reconstruction error, our model performs on par with Wonder3D and has slightly larger standard deviation. This is expected since our model is designed to avoid overfitting to its training set and to correctly cover a diverse solution space instead of returning one solution.
>
> Please also revisit Appendix Table 1, which shows the quantitative results for top-5 outputs and comparisons to additional baselines.
>
> **Questions**:
> > Since the method yields multiple solutions satisfying concave/convex ambiguities. I am wondering if it is possible to determine the exact number of solutions. For example, in figure 5, 3 solutions are reported. However, it is easy to see that there are only 2 solutions possible in these cases satisfying the discrete (convex/concave ambiguity are possible. The third solution is normally a bas-relief of one of these two solution. Therefore, it would be nice to add a mechanism to classify the solutions according to possible ambiguities.
>
> Yes, one would think that the solution space usually consists of globally convex/concave reconstructions, each with their own family of GBR variations. However, the results from [29] suggest that regions enclosed by so-called “critical contours” of shading can each flip independently, giving even more solutions. An example adapted from [29] is shown in Appendix Fig.9b (nested rings), where we see that our method successfully recovers more solutions.
>
> When global lighting consistency is included, some solutions are eliminated by our algorithm, typically leaving two main modes that have a global convex/concave relationship. We believe it is important to maintain flexibility in how and when global lighting consistency is applied, because excluding it allows exploring a larger solution space, and because perceptual studies like [*] include compelling examples that make it clear that lighting consistency at the global level is not always strictly enforced in human perception.
>
> To characterize the outputs of our model for a specific image, we find it helpful to inspect a lower-dimensional projection, such as from t-SNE, to assess the presence of discernible clusters. Figure 6 shows an example.
>
> We thank the reviewer for this question, which will help clarify these points in the final revision.
>
> * [29] Benjamin Kunsberg and Steven W Zucker. From boundaries to bumps: when closed (extremal) contours are critical. Journal of Vision, 21(13):7–7, 2021.
>
> * [*] Yuri Ostrovsky, Patrick Cavanagh, and Pawan Sinha. Perceiving illumination inconsistencies in scenes. Perception, 34(11):1301–1314, 2005.

---

> > ### Comment · Reviewer_xHhf · 2024-08-12
> >
> > I thank the authors for the rebuttal.
> >
> > I don't have any questions at the moment.

---

### Author Rebuttal · Authors · 2024-08-06

We thank all reviewers for their thoughtful comments. We are encouraged that all reviewers find our research problem interesting and our lightweight patch diffusion model novel and effective. Reviewers (3BEx) and (E5e1) also find our presentation to be well-written and reviewer (xHhf) finds the ablation studies helpful.

Here we address common questions from all reviewers: What problems can benefit from the proposed model, and how will it be useful in practice?

Multistable perception happens much more often than one may imagine. When visual cues that help resolve the shape are missing, such as occluding boundaries, depth profiles can become ambiguous, as clearly demonstrated in our figures. One may not often notice this in their daily life because humans usually have access to additional visual cues. Yet, it shouldn’t be hard to recall an occasion when your shape perception was corrected by conscious reasoning or action—such as consciously considering the context of occlusions and lighting, or by moving your head to gather additional viewpoints—after an incorrect first glance at a scene. Models like ours are essential for understanding how human perception achieves this fine balance of efficiency and robustness, and how it can be imitated in robotics.

As for relevance to scientific disciplines beyond human and robot perception, consider astronomical imaging. Our attached PDF includes two photographs of the Martian surface. Humans misperceive a crater as a mountain and vice versa in these photos, due to their ingrained perceptual bias. (These are instances of the so-called crater illusion.) Data acquisition is expensive in these situations, so there is benefit to having a monocular vision model that can elucidate the possibility of multiple solutions, thereby allowing all possibilities to be examined by gathering additional context. Similar to human perception, in astronomical imaging it is essential to have access to all possible “bottom-up” solutions, so that one can use context or “top-down” information as effectively as possible.

As you can see in the attached **PDF Fig. R1**, we tested our model on two crater images, along with two other diffusion-based models and the Depth Anything model. Our model recognizes both possibilities, crater and mountain, while the other models only see one of the two.

We will adjust the text in our paper to clarify these important elements of our model, in the broad context of visual perception.

---

### Decision · Program_Chairs · 2024-09-25

**Decision:**

Accept (spotlight)

**Comment:**

This paper addresses a classical problem in computer vision: shape from shading. It demonstrates that it is possible to recover "multistable" shapes from a single image using a diffusion-like process. All reviewers have recommended acceptance, with one reviewer giving a strong acceptance. The AC concurs with the reviewers' assessments and strongly recommends acceptance.